# User Engagement Comparison between Advergames and Traditional Advertising Using EEG: Does the User's Engagement Influence Purchase Intention?

Ivonne Angelica Castiblanco Jimenez *, Juan Sebastian Gomez Acevedo, Elena Carlotta Olivetti, Federica Marcolin *, Luca Ulrich, Sandro Moos and Enrico Vezzetti

Department of Management and Production Engineering, Politecnico di Torino, Corso Duca degli Abruzzi 24, 10129 Turin, Italy
* Correspondence: ivonne.castiblanco@polito.it (I.A.C.J.); federica.marcolin@polito.it (F.M.)

**Abstract:** In the context of human–computer interaction (HCI), understanding user engagement (UE) while interacting with a product or service can provide valuable information for enhancing the design process. UE has been a priority research theme within HCI, as it assesses the user experience by studying the individual's behavioral response to some stimulus. Many studies looking to quantify the UE are available; however, most use self-report methods that rely only on participants' answers. This study aims to explore a non-traditional method, specifically electroencephalography, to analyze users' engagement while interacting with an advergame, an interactive form of advertising in video games. We aim to understand if a more interactive type of advertising will enhance the UE and whether, at the same time, it would influence the user's purchase intention (UPI). To do this, we computed and compared the UE during the interaction with an advergame and a conventional TV commercial while measuring the participants' brain activity. After the interaction with both types of advertising, the UPI was also evaluated. The findings demonstrate that a more interactive advertisement increased the participants' UE and that, in most cases, a UE increment positively influenced the UPI. This study shows an example of the potential of physiological feedback applications to explore the users' perceptions during and after the human–product interaction. The findings show how physiological methods can be used along with traditional ones for enhancing the UE analysis and provide helpful information about the advantages of engagement measurement in HCI applications.

**Keywords:** user engagement; EEG; purchase intention; advergames

## 1. Introduction

With an increasing number of digital services, HCI studies regarding the analysis of interfaces and approaches to exchange information are a must. In particular, exploring innovative techniques that allow researchers and practitioners to understand users' needs without bias may foster HCI applications in promising consumer–content interaction fields. Affective computing developments have been a matter of interest for interdisciplinary fields due to the practical implications concerning the study and design of systems capable of recognizing and interpreting human emotions. The marketing and video games industries are no exception. Some years ago, video games were a minor activity exclusively earmarked for kids, leaving aside the adult audience. The actual situation of video games is undoubtedly much different: video games ranked first in leisure activities, surpassing the traditional ones such as cinema and music [1]. Additionally, their importance continues to increase, reaching a media resonance and social impact due to the development and implementation of new technologies, especially in an era affected by the COVID-19 pandemic, where technology has taken on a significant role in the entertainment market in terms of popularity and profitability [2]. The impact of the ongoing pandemic, social restrictions,

and the recommendation to stay at home have significantly contributed to a rise in the gaming industry's numbers of users.

In an environment where consumers have the choice to pay and get what they desire, advertising can be quickly rejected, and the impact of video games has not gone unnoticed by the publicity market, a sector reinventing itself to find new ways to reach an even more demanding audience. The advergames, a hybrid between advertising and entertainment content in video games, can contribute to a better perception of advertising by developing new possibilities for marketing communication. Advergames have also gained a competitive advantage [3] for attracting and engaging users due to their interactivity and high exposure to a brand or product [4] while offering a non-intrusive experience. The user is the one who addresses the game and interacts with it voluntarily. Additionally, this type of interactive advertising allows easy customization since the brand can include its style and values to create a unique and representative product, thus contributing to the brand's personality and attracting more users who identify with the brand.

This effectiveness is achievable thanks to the capacity of introducing advertising content inside an environment where the users have shown a good perception of the publicity. Furthermore, the value of advergames lies in their ability to provide a compelling advertising message that consumers are more willing to accept [5]. Due to these advantages and the significant impact that video games have worldwide, advertising campaigns should consider advergames, a non-saturated leisure platform [6], as an opportunity to increase user engagement (UE) and contribute to user purchase intention (UPI). In other words, advergames could take advantage of the UE with video games to reach a more demanding audience and increase the willingness to acquire a product.

UE, a typical human–computer interaction parameter, has been described as the degree of focus and immersion in a particular task [7]. Traditionally, this indicator has been measured using almost exclusively self-report measures; however, in order not to rely only on respondents' answers, technological advances have made possible the use of new instruments that can provide physiological outcomes to examine the "ground truth" of the users' perceptions. Among these, electroencephalography (EEG), a non-invasive procedure for measuring and examining the electrical activity of the brain, is becoming a potential instrument for exploring UE. Indeed, by analyzing EEG data collected while participants were exposed to a stimulus, McMahan and Freeman's studies [8,9] have demonstrated that EEG-based metrics can be used to determine whether a user is more or less engaged. The results from the literature would suppose that an advergame will provoke higher engagement than a TV commercial since TV is a passive media; however, it is unclear if a more interactive type of advertising would lead to a higher UPI. The goal of meaningfully quantifying the UE using physiological tools such as EEG will provide additional helpful information that, combined with traditional methods, can enrich this research field.

This empirical study aims to validate the hypothesis that "a more interactive type of advertising impacts the UE, positively influencing the UPI". To do this, we have created an experiment where the users are exposed to two advertisements for the same product whilst their brains' activity is being recorded. In this way, we aim to compare a traditional advertising campaign with an advergame to quantify the UE difference using EEG. A validated questionnaire will be used to determine the UPI before and after the experiment.

This paper is structured as follows: Section 2 presents the research background, including a review of the UE in video games, the EEG as a method for understanding the UE, and the conceptual relation between UE and UPI. Section 3 presents the methodology, the experimental setup, and the data analysis. Section 4 presents the analysis results, and Sections 5 and 6 contain the discussion and conclusions.

## 2. Research Background

UE has been a matter of interest in different fields, including education, healthcare, and entertainment. The way UE has been evaluated differs from study to study. For the purpose of this research, we will focus our attention on the most common methods to

measure the UE during video game interactions and the exploration of EEG as a means to measure this indicator.

### 2.1. An Overview of User Engagement in Video Games

UE can be defined as the absorption level and willingness to execute a task [7]. In a recent study, Kniestedt et al. [10] delineated the engagement definition in applied games as a concept involving the state of concentrating attention on a task and the involvement with the context where that task occurs. The UE in video games has been a matter of interest in different studies [11–16] under the idea that users can be engaged in activities through meaningful interactions. The interactive possibilities that video game technologies offer can enhance engagement more than other activities. Brockmyer et al. [12] designed the Game Engagement Questionnaire (GEQ) focused on analyzing video games' violence. The study explores the engagement experience in four dimensions: absorption, flow, presence, and immersion. It concludes that it is possible to identify the users' psychological engagement when playing video games and that a higher UE can significantly impact game playing.

On the other hand, Bianchi-Berthouze [14] considered how body movement could affect the UE during gameplay. The movement-based engagement model specifies the relationship between movements and engagement. The model classifies the body movements into five groups and describes their relationships with the engagement. The study concludes that the freedom offered by video games plays an essential role in the UE, allowing a more in-depth immersion in the scenarios presented, attracting users to continue exploring and engaging more and more with the game. This work has been widely appreciated by the scientific community and is an excellent example of the importance of physiological measurements to better understand the UE.

Another significant study is the one carried out by Martey et al. [15], who measured the UE with video games, measuring different characteristics that may influence the users' perception. These characteristics are a player avatar, visual realism, and narrative. In this study, the authors measured five indicators (a self-report survey, the attention to the stimulus, Galvanic skin response (GSR), mouse movements, and game click logs) when playing a video game to measure how these characteristics can influence the capacity of a video game to engage the user. The results indicated that physiological measurements (GSR, mouse movement, and clicks) are highly related to users' expressions and feelings and can provide reliable information for UE identification and analysis.

Abdul Jabbar and Felicia [16] developed a model where emotions and thoughts play an essential role in UE and learning. The study illustrated the impact of some gaming features on cognitive and emotional levels by classifying the engaging elements into four types: motivational, interactive, fun, and multimedia. The study found that some aspects that might influence the UE in the gameplay are related to visuals, role-play, obstacles, virtual environments, control/choice, and rewards.

Likewise, O'Brien and Toms [17] created a survey to test the UE of software applications, named the User Engagement Scale (UES). In this survey, users are asked to answer questions about their experience with the product. This survey was later adapted by Landa-Avila and Cruz [18] for a virtual game to analyze six aspects: endurability, novelty, perceived usability, aesthetics, felt involvement, and focused attention. After data analysis, the authors found that the UES questionnaire can be used to measure the UE in a video games context; however, there were objections about the test's duration and misunderstandings about some items. This situation was later examined by O'Brien et al. [19], who proposed the UES short form.

Other studies on UE analysis in video games are listed in Table 1. Most of these studies, however, relied only on self-report methods, such as questionnaires, interviews, surveys, and scales, in which participants were asked to answer questions or provide responses to prompts that are designed to collect data on a particular topic or construct. The data collected through self-reported methods can be subjective and may be influenced by various factors, such as the participants' memory, motivation, and willingness to reveal

personal information. Despite these limitations, self-reports have been widely used as a method for UE analysis, and consequently, this could lead to biased outcomes due to the nature of the responses. In this sense, self-reports could be compared and enhanced with other experimental approaches that do not entirely depend on participants' responses, such as the case of EEG.

**Table 1.** UE analysis in video games.

| Author | Method | Purpose/Outcome |
| --- | --- | --- |
| Mayes and Cotton [20] | Questionnaire | Development of an engagement questionnaire (EQ) in video games |
| Lankoski [21] | Framework | Engagement definition classified as goal-related or empathic. |
| Schoenau-Fog [22] | Survey | Definition of a player engagement framework based on objectives, activities, accomplish, and affect. |
| Schonau-Fog and Bjorner [23] | Interview | Method to classify the experience of engagement with video games into six types: sensory, intellectual, physical, social, narrative, and emotional engagement. |
| Corem et al. [24] | Scale | Proposal of a player-matching system based on the Elo rating system. |
| Sharek and Wieve [25] | Physiological (haptic) and Survey | Engagement measurement based on how many times a participant clicked a game-clock. |
| Wieve et al. [26] | Scale | A modified version of the User Engagement Scale (UES) for video game research. |
| Kirschner and Williams [27] | Interview | Gameplay review method (GRM) for players' engagement using interviews and audiovisual recordings |
| Pope et al. [28] | Physiological (EEG) | EEG engagement index based on electroencephalographic signals. |
| McMahan et al. [8] | Physiological (EEG) | Identification and verification of the EEG engagement index developed by Pope et al. [28] as a valid indicator of UE with video games. |
| Phan et al. [29] | Scale | Game User Experience Satisfaction Scale (GUESS) |
| Sreejesh and Anusree [30] | Survey | Study on the effects of cognition demand and brand attention. |
| Sawyer et al. [31] | Scale | Hierarchical Bayesian method to model player engagement. |
| Nermend and Duda [32] | Physiological (eye tracking, GSR) and Questionnaire | Methodology for choosing the location of advertising in games, using eye tracker and GSR. |
| Barclay and Bowers [33] | Survey | Use of the Revised Game Engagement Model (R-GEM) to conclude that immersion in an activity may not only be a result of extremely usable systems or particularly receptive users but an effect of the experience itself. |

### 2.2. EEG as a Means for Understanding the Engagement

The literature on the methods, measures, and elements that influence the UE with video games shows a predominance of self-report methods; however, methods based on physiological analysis are becoming more accessible due to the technological advances that have increased the availability and usability of these systems. An example is a study conducted by McMahan et al. [8]. In this research, the authors used EEG to evaluate the

player task engagement in video games in two situations, during normal gameplay and player death. The study identified that the engagement corresponds to immersion states, concluding that higher levels of engagement are present during death events compared to general play. Likewise, a recent study [34] used an educational video game created to test medical students' clinical skills and track their engagement using EEG. The results showed a correspondence between the EEG outcomes and the participants' performance.

The evaluation of engagement with EEG has also been explored in the education field. Chaouachi and Frasson [35] analyzed the effect of engagement on the students and found that it strongly influences the response time and can be an indicator of learners' performance. At the same time, the authors used the engagement index, initially proposed by Pope et al. [28] and later validated by Freeman et al. [9]. Similarly, Coelli et al. [36] evaluated the relationship between the level of cognitive engagement and focused attention, concluding that EEG can be used to detect changes in mental state and subjects' reaction times. A recent study [37] proposed a device prototype consisting of an EEG headset and a scarf that provide haptic feedback when a decrease in attention is detected.

Considering the emotional research, Ramirez et al. [38] conducted a study based on EEG to analyze the emotional effect of music in advanced cancer patients, and their findings showed a decrease in anxiety and tiredness. Likewise, in a previous study, Giraldo and Ramirez [39] described an approach to calculate arousal and valence values from EEG activity in real-time. On the other hand, Ramirez and Vamvakousis et al. [39] explored a machine learning approach to detect emotions from EEG signals. In these studies, the authors computed and validated EEG-based indices for obtaining information about the users' emotional states.

A further study [40] explored the engagement analysis with advertising. This study focused on traditional advertising and did not use a validated engagement EEG index, and the EEG data were globally recorded and then correlated with the participant's answers to a questionnaire. However, the results were not conclusive, and the authors recognized the need for more profound research in this field. Likewise, other studies [41] have focused their interest on analyzing players' emotions in VR gaming using EEG data and comparing their results with self-report methods.

The studies on engagement and EEG showed that EEG-based metrics are suitable for studying UE. However, despite the advances in the study of engagement in advertising and video games, these two fields have been explored individually; consequently, there is a research gap regarding the engagement during advergames interactions using EEG.

### 2.3. Relation between Engagement and User Purchase Intention in Advergames

UPI can be defined as the conscious decision of a user to buy a particular product [42]. This degree of willingness to pay is a variable that depends on several factors, for example, product perception, product price, attitude toward a specific purchase, the interaction experience, engagement with a particular product or brand, and customer service. Survey instruments have been a common choice for assessing these variables and ultimately estimating the UPI.

Goh and Ping [11] developed and validated a post-game questionnaire to measure the UPI and attitude toward the advergame experience, the brand, and ads in general. The study concluded that the attitude toward the brand positively influences the UPI and confirms the advergames' added value in effectively increasing the UPI of the advertised product.

Chen [43] developed a model of the in-game factors influencing the user purchase reaction with an advergame. The study identified that audiences are more likely to develop acquisition desire and positive brand attitudes when they are engaged and attracted during the advergames. On the other hand, Catalan et al. [44] identified a group of elements that could affect user experience and purchase reaction in a mobile advergame. The study defined the following factors that might influence the UPI: skills, challenge, interactivity, focused attention (engagement), and telepresence.

Other studies concluded that advergames lead to a higher purchase desire than TV commercials [45], that UPI and attitude towards a brand increase when there are elements of interactivity [46], and that children who are more involved and have a positive attitude during the game show a higher UPI [47].

Finally, Chang et al. [48] proposed a conceptual framework for exploring the congruity, integration, and prominence as variables persuading the interest and UPI during in-game advertising. The study suggested a survey instrument for measuring the UPI and concluded that user attention/engagement positively affects the purchase interest during gameplay.

Considering the previous findings that highlighted UE as one of the most significant variables that could influence the customer's desire to acquire, in this research, the UE influence on the UPI during an advergame interaction and projection of a tv commercial has also been analyzed.

## 3. Methods and Tools

The relationship between video games and advertising is complex and multifaceted, with each industry influencing and benefiting from the other in various ways. In the intersection between, there are the advergames (Figure 1), which provide a unique and highly engaging platform for advertisers to reach consumers, with in-game ads and sponsored content offering new and innovative ways for brands to connect with their audience and promote a product or service through the use of interactive and engaging gameplay. This strategy relies on user engagement to be successful, with the level of engagement (UE) influencing the effectiveness of the advertising and the likelihood of a user making a purchase (UPI).

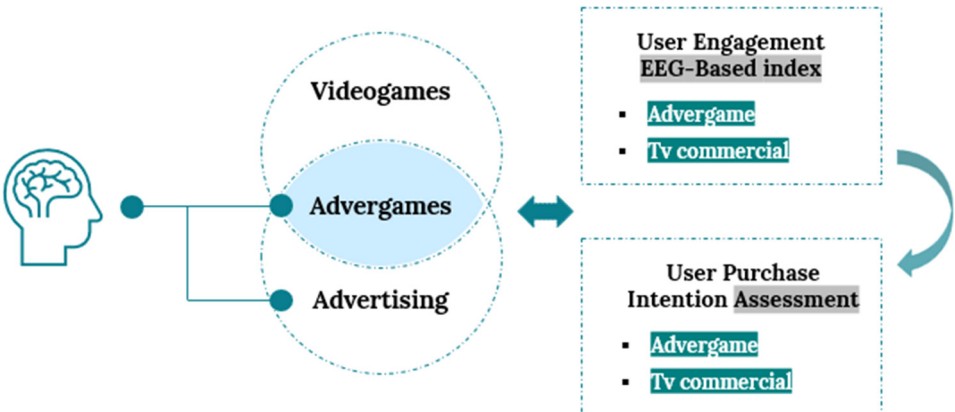

**Figure 1.** Model of UE influence on UPI.

The interaction between these three contexts (video games, advertising, and advergames) and the UE influence on the UPI frames this research.

This research aims to measure the UE with an advergame and a traditional advertising campaign to analyze if the UE has an incidence on the UPI of a product or brand exposed. To achieve this, we focused the literature background analysis on previous UE and video games research, EEG as a method for UE identification, and the UE influence on the UPI in advergames.

The literature review shows that no study on advergames has been conducted using EEG as a method to measure the UE and its influence on the UPI. Besides, we want to contribute to this field by endorsing previously validated tools employed individually in related areas (games and advertising) to verify the hypothesis: "A more interactive type of advertising impacts the user perception and engagement, positively influencing the purchase intention".

To verify the previous hypothesis, the participants' physiological reactions were measured using EEG to analyze the UE during an advergame and a TV commercial. After both advertising interactions, the UPI was calculated using a questionnaire assessment.

### 3.1. EEG Engagement Index Data Analysis and Feature Extraction

The neuronal electrical activity is composed of different frequencies that can be obtained through the EEG signal, and these brain waves are grouped into main frequency bands, such as: theta ($\theta$), alpha ($\alpha$), beta ($\beta$), and gamma ($\gamma$). Theta waves have a frequency of 4–8 Hz, alpha 8–12 Hz, beta 12–25 Hz, and gamma 25–45 Hz. These different frequency bands are commonly associated with diverse cognitive processes [35].

Beta ($\beta$) power is related to system activation and higher mental activity of the brain when a person is aware and cognitively engaged. In contrast, alpha ($\alpha$) is associated with lower mental vigilance and usually appears in sleep–wake cycles; at the same time, theta ($\theta$) activity occurs most often in sleep or deep meditative states. Pope et al. [28] proposed an EEG-based index to analyze UE; in their research, the authors assessed three possible indices computed using the EEG frequencies' bandwidth and suggested the most accurate UE index (Equation (1)):

$$Engagement\ Index = \frac{\beta}{\alpha + \theta} \tag{1}$$

Following this approach, the UE index (1) considers the ratio between the beta waves (higher mental activity of the brain) and the sum of the theta and alpha waves (both related to lower mental activity of the brain).

This research was carried out employing a mobile EEG headset (Emotiv EPOC+) with 14 wet-EEG electrodes (AF3, AF4, F3, F4, F7, F8, FC5, FC6, P7, P8, T7, T8, O1, O2) placed at standard 10–20 positions (Figure 2). Two additional CMS/DRL reference channels were located at P3 and P4. The sampling rate was 128 Hz, and the bandwidth was between 0.2 and 45 Hz. The band power calculation was performed at 0.125 s intervals (8 Hz). We applied a FFT to the most recent 2 s epoch of EEG data and averaged the FFT-squared magnitude across the frequencies in each band. Before performing the calculation, the EEG stream passed through a 0.18 Hz high-pass FIR filter. To improve the accuracy of the spectral analysis and reduce artifacts from the procedure, a Hanning window was implemented. For the engagement calculation, Equation (1) was employed. Similarly to [49], the engagement index (1) for each participant was computed considering the averaged overall electrodes' measurements.

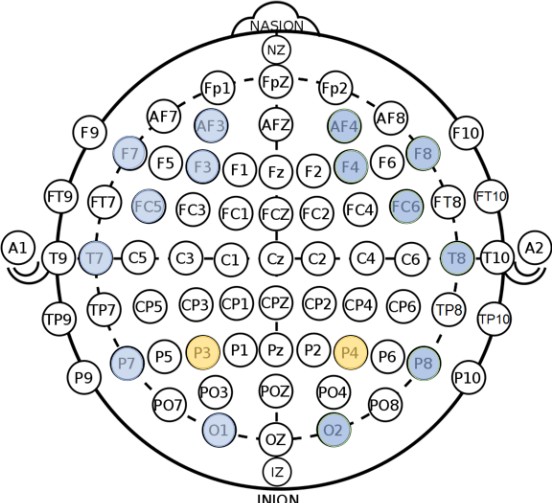

**Figure 2.** Sensor location. The electrodes' positions employed in this study are highlighted.

Considering that in physiological studies, the initial baseline and the subsequent responses (after stimulus) are highly individual-dependent and differ from participant to participant, there are no absolute (max or min) values for UE. To help reduce errors associated with individual differences, this research employed a "within-subject design" experiment, often used in similar studies [8,9,35,49], to regulate the subjects' measurements by exposing them to all experimental conditions. Users bring their own background and

current physical and mental state to the test. Therefore, to minimize the random noise, the EEG data were separated into two phases: The first contains the baseline recording during a relaxed period, useful to identify the primary user's state, and used as a reference point for comparing the brain's activity during different tasks. The second contains the electrical brain data while exposed to a stimulus, in this case, while interacting with the advergame and while watching the traditional advertising. By analyzing and contrasting the brain activity difference between the baseline and the activated engagement during the stimuli, it is possible to identify the activated response and, thus, the increment/decrement of the engagement index; if it increases, this can be interpreted as a higher level of UE (Figure 3).

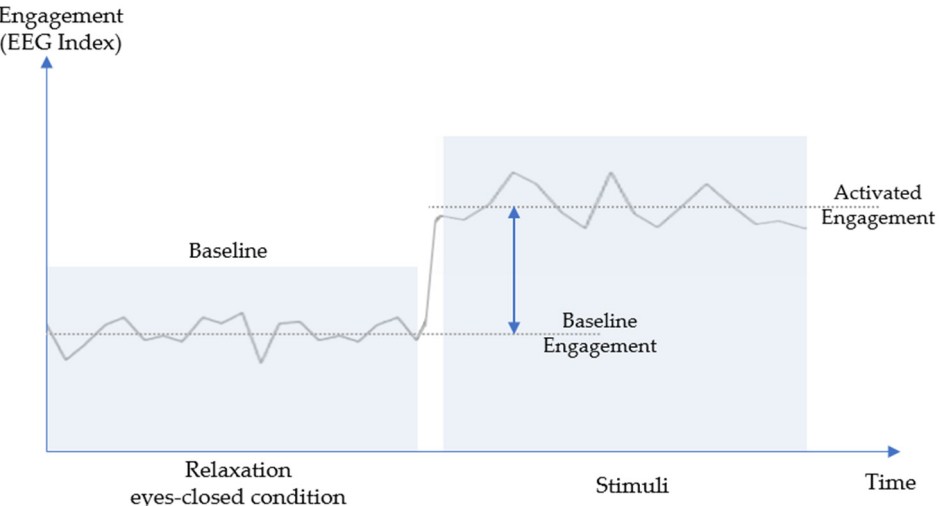

**Figure 3.** Engagement baseline vs. stimuli activation.

In this way, the participants in a within-subjects design serve as their own control by providing baseline metrics across the different stimuli. After analyzing and normalizing the entire signals, it is possible to generate a general comparable data analysis for all the participants exposed to the same experiment. The goal is to measure the variation resulting from the different stimuli for the engagement outcomes.

*3.2. Experiment Setup*

A controlled experiment (Figure 4) was conducted in a lab environment with 24 healthy participants, 13 women (55%) and 11 men (45%), aged between 23 and 45, recruited in person. All of them reported computer skills. All participants signed an informed consent after being briefed about the experiment, the use of their data, and its protection. Then, each user was accompanied to perform the test in a controlled environment, a closed room equipped with an adjustable chair, a display monitor, and a gamepad, planned to avoid situations that may influence the results. The experiment was displayed on a 40-inch curved display with an effective viewing area of 884.74 mm (horizontal) and 497.66 mm (vertical) at a 3000 mm radius curvature with a pixel density of 110 per inch. The participants sat approximately 800 mm (±50 mm) in front of the display. This monitor was selected as studies [50,51] showed that considering a large field of view of a participant can greatly enhance their experience, and with the help of its curvature and high pixel density, it can increase immersion.

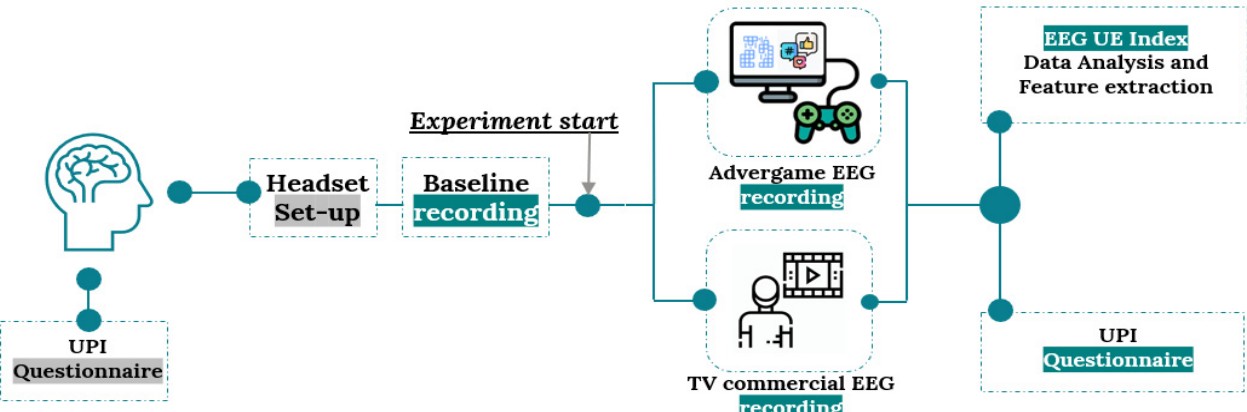

**Figure 4.** Experiment setup.

First, the users were requested to fill in the UPI questionnaire according to their original perception of the product selected for the study. The EEG headset was positioned on the participant's head, and the correct connection of the electrodes was verified. The participant was invited to relax for 3 min. After the training phase and setting up, to facilitate the immersion and successfully conduct the experiment, the lights were turned off.

A baseline period of 30 s was recorded during the eyes-closed condition. Half of the participants were randomly assigned first to play an advergame and then proceed with a TV commercial, while the other half initially watched the TV commercial and then continued with the advergame activity. In both cases, the participants were asked to play the advergame for 4 min and exposed to a TV commercial for 3 min while measuring their brain activity. Between one activity and the other, the participants were invited to relax for 2 min. At the end of each type of advertising, the participants were requested to fill in the UPI questionnaire again.

To evaluate the UPI, we used the purchase intention evaluation proposed by Chang et al. [48] to assess the consideration, desire, plan, and likelihood of buying the advertised product using a seven-point Likert scale.

The selected product belongs to a well-known brand of carbonated soft drinks. The TV commercial displays an urban scenario with young people singing and dancing in the street while inviting the spectators to join their march, the melody set in the background recalls the brand's iconic music. The advergame is an endless runner third-person perspective game developed to avoid obstacles by running, dashing, and jumping, and the user's role is to save dehydrated people by collecting and delivering them cans of drink. The atmosphere is designed to show branded ads blended with the surroundings throughout the game.

## 4. Results

The raw EEG data were collected and processed. Artifacts due to body movements and signal noise were removed. Twenty results from the total sample were deemed suitable for the objectives of this study. The EEG data were segmented into the baseline (eyes-closed condition) and activity periods when interacting with the advergame or watching the TV commercial. To determine and contrast the EEG engagement index between periods, the baseline data were compared to the post-stimulus response for both the advergame and the TV commercial; in this way, it was possible to examine the influence of the advertising on the UE.

The EEG engagement index average for each one of the participants is shown in Figure 5, divided into the baseline and activity periods. The results show that during the advergame interaction, the engagement was always higher than the baseline period, indicating increased attention and vigilance while gaming.

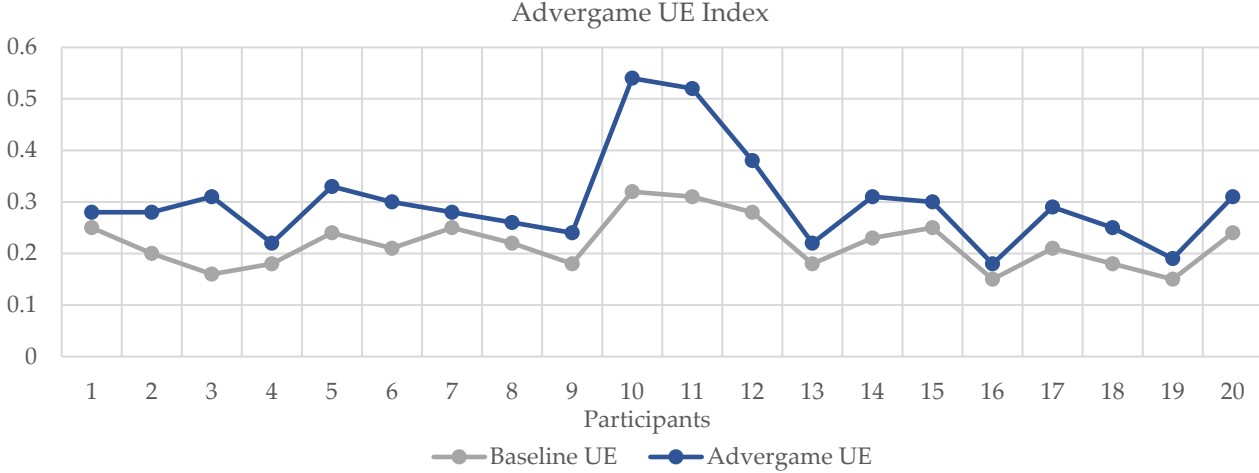

**Figure 5.** UE index during the advergame interaction.

Regarding the TV commercial, the results show (Figure 6) that in some cases, the EEG engagement index was lower than the baseline period, thus showing a decrease in attention and vigilance, in other words, a lower UE.

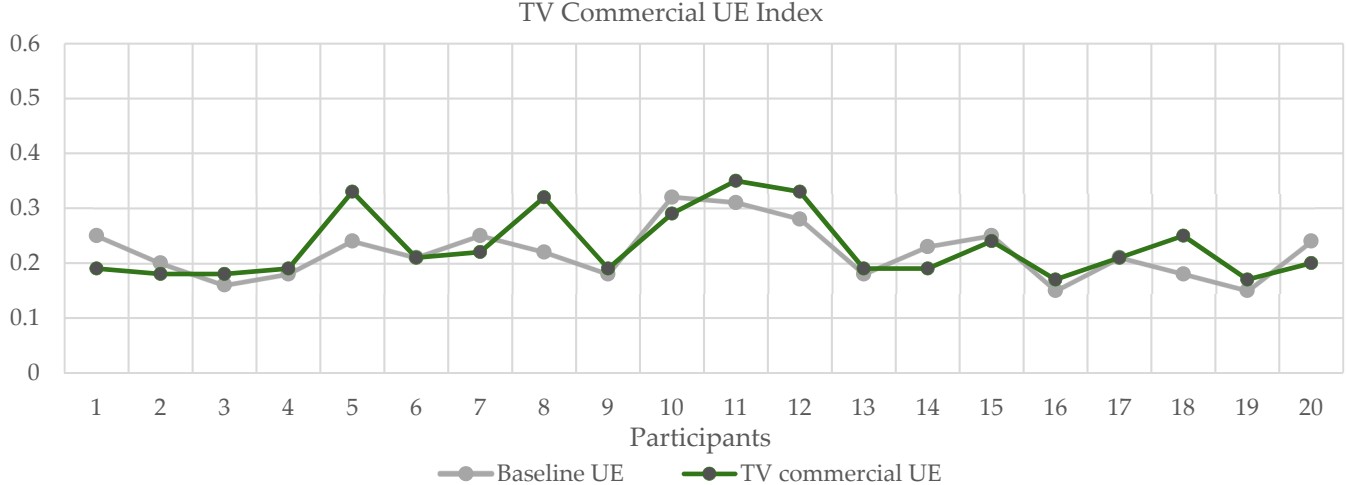

**Figure 6.** UE index during the TV commercial.

The average UPI for each advertisement and participant is shown in Figure 7. In the case of the TV commercial, the UPI was higher for only 30% of the participants compared to the original UPI reported at the beginning of the experiment. For the advergame, the results show that the UPI was usually higher (70% of the cases) than the original UPI before the advertising interactions. This indicates that advergames generate a post-game response that tends to positively influence the purchase intention/reaction of the participants, encouraging them to consume the product.

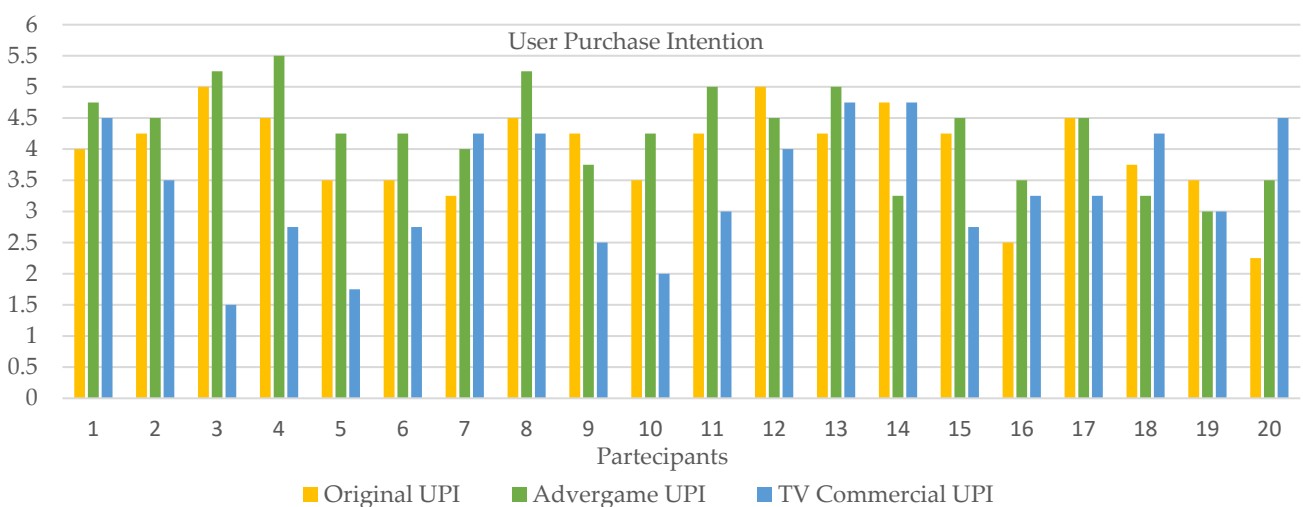

**Figure 7.** UPI before and after the advertising interactions.

In some cases, however (participants 7, 14, 18, and 20), the UPI was higher after seeing the commercial in contrast to the original UPI and advergame, indicating that for a minority of participants, the traditional advertising has a higher incidence on the UPI. Additionally, it is interesting to notice that for three participants (9, 12, 19), the original UPI was higher than the UPI reported after the advergame and TV commercial activity, representing a negative influence of both advertisements. Thus, considering the total contribution of both advertisements, 25% of the participants reported a positive UPI response for both types (advergame and TV commercial), while 15% reported a negative response. Advergames overall generated the most significant positive responses from the stimulus.

## 5. Discussion

The results support the initial hypothesis: "A more interactive type of advertising impacts the user perception and engagement, positively influencing the purchase intention". An analysis of variance (ANOVA) on the EEG engagement index was performed to verify the statistical difference between the EEG engagement index while playing an advergame vs. seeing the TV commercial, obtaining: ($F_{(1,38)} = 7.40$, $p < 0.01$). The outcomes show that a more engaging type of advertising significantly influences the UE; in this case, an advergame had a more significant influence on the UE than a traditional TV commercial. This result indicates that the EEG engagement index was statistically significantly higher when the participants interacted with the advergames than when they watched the TV commercial.

Still, the fact that a stimulus produces a more engaging behavior would not necessarily imply that the UE will reflect similarly on the UPI. Hence, to analyze if a more interactive type of advertising generates a more significant influence on purchase intention, an ANOVA was also performed for the average UPI of the participants for both activities, obtaining: ($F_{(1,38)} = 10.75$, $p < 0.01$). The previous result showed that after playing the advergame, the UPI was statistically significantly higher than the UPI after seeing the TV commercial.

The interactions between UE and UPI can be distributed in four clusters according to the participants' responses. For the context of this study, we have developed a four-quadrant model (Figure 8) to show the classification considering the reported UPI and the EEG engagement index variation.

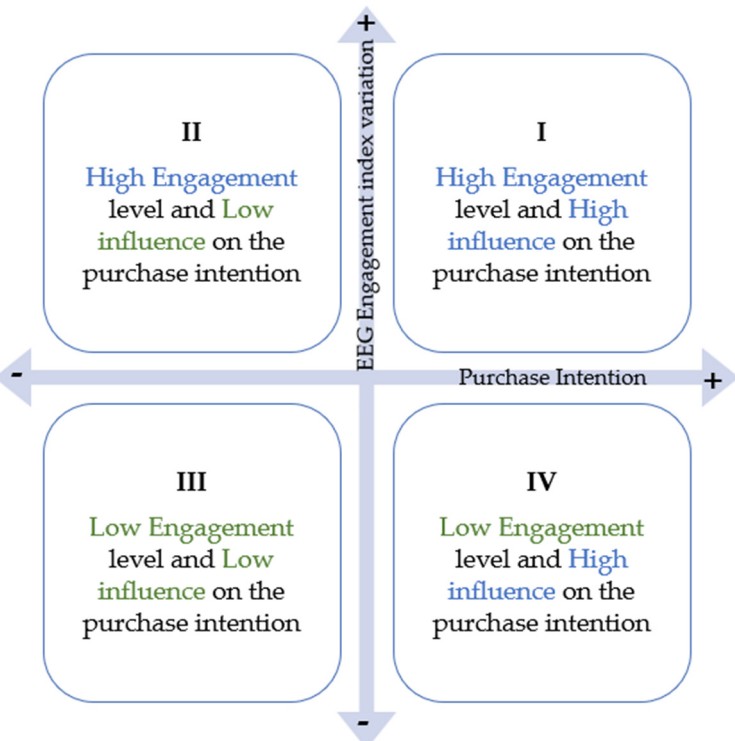

**Figure 8.** Quadrant model of EEG engagement index variation vs. UPI.

The X-axis refers to the UPI. Number 4 is located at the center since it indicates the neutral response of the participants in the questionnaire. The Y-axis shows the difference between the EEG engagement index during the stimuli and the baseline. By contrasting the difference between the resting period (baseline) and the advertising interaction during the test, it is possible to identify the variation of the index; in other words, if the variation is positive, the stimuli produced an increase in the engagement, and vice versa.

For the advergame (Figure 9), all the participants presented a positive variation in the EEG engagement index, indicating an increase in the UE when interacting with it. In the first quadrant, there were thirteen participants (65%), in the second quadrant, there were seven participants (35%), and in the third and fourth quadrants, there were no participants.

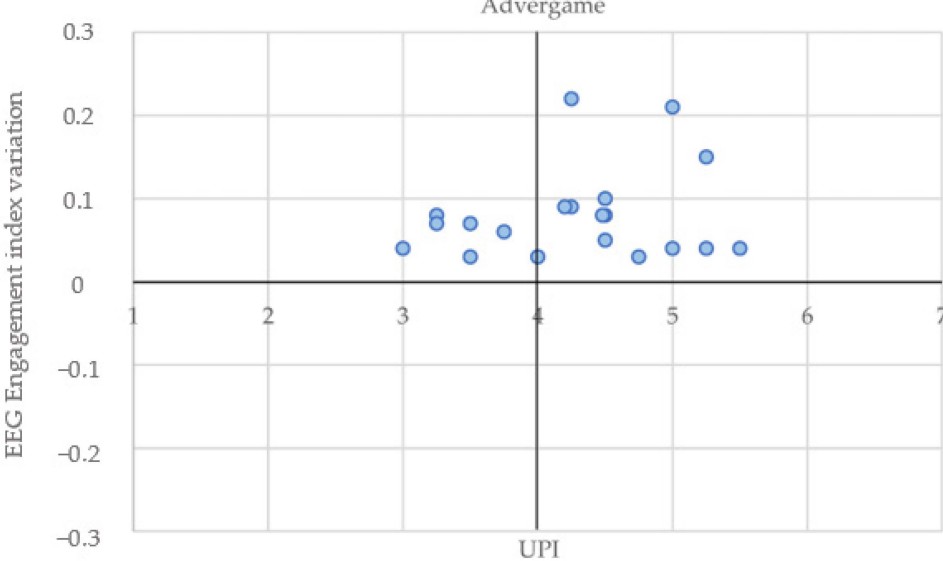

**Figure 9.** Advergame—Quadrant model of EEG engagement index variation vs. UPI.

For the TV commercial (Figure 10), positive and negative changes in the index were identified, which means that in some cases, there was a negative variation of the EEG engagement index, and in other cases, there was a positive variation. In the first quadrant, there were three participants (15%), in the second quadrant, there were ten participants (50%), in the third quadrant, there were five participants (25%), and in the fourth quadrant, there were four participants (20%).

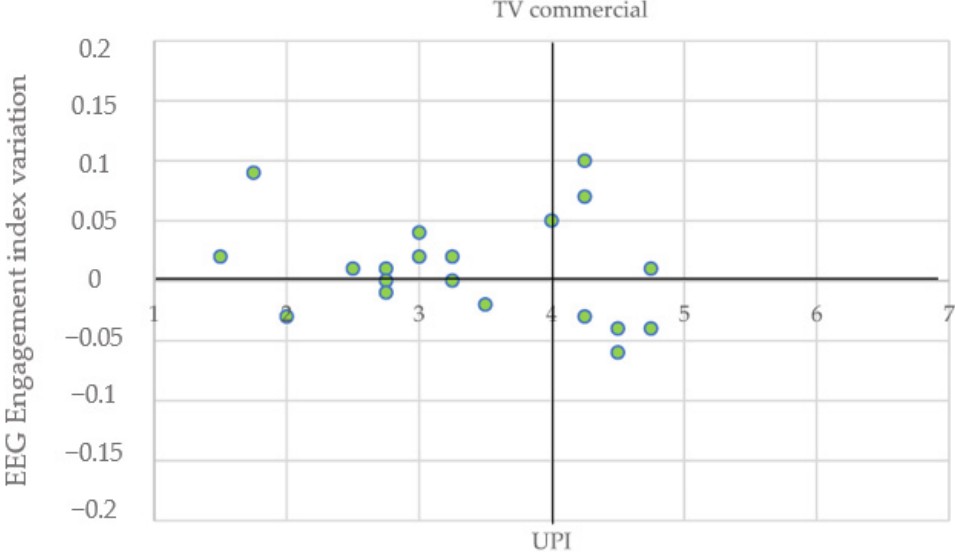

**Figure 10.** TV commercial—Quadrant model of EEG engagement index variation vs. UPI.

The results in Figure 10 confirm a decrease in the number of participants located in the first quadrant for the TV commercial compared to the results obtained in the advergame (Figure 9), indicating that a lower number of participants presented a high level of engagement and a high UPI while watching the traditional advertisement. In the second quadrant, there was a similarity in the number of participants between the TV commercial and the advergame, indicating a high level of engagement and a low UPI. This result suggests that even if the participant was engaged, the advertising did not affect the UPI of these participants. Only the TV commercial had participants in the third quadrant, showing that traditional advertising is sometimes not attractive. In the fourth quadrant, the TV commercial involved four participants who experienced a decrease in the EEG engagement index but still reported a positive UPI. These persons located in the fourth quadrant reported a positive attitude toward the brand at the start of the test, confirming the study performed by Nelson et al. [52], who indicated that users that have a positive attitude toward the brand or product advertised in games have a higher purchase predisposition.

Finally, the contrast in the total number of points in the rightmost halves of both Figures 9 and 10 shows that the advergame had a positive impact on UPI for thirteen participants (65%), while the TV commercial only for seven (35%). This implies that apart from the UE elicitation, the advergame was most effective at increasing participants' purchasing interest. By playing the advergame, users were exposed to the product in a pleasant and enjoyable way, enhancing their interest in it and increasing their likelihood of considering buying it. Overall, advergames can be an effective marketing strategy to inform customers about the product and its characteristics, improve users' understanding of the product, and boost their desire to purchase it.

## 6. Conclusions

This study aimed to explore engagement and its influence on purchase intention using a physiological tool, specifically electroencephalography. According to the literature, most studies implemented self-report methods to identify the relationship between video games

and UE. Additionally, no previous studies have analyzed the UE during an advergame and its implications on the UPI compared to traditional advertising using EEG.

The study's objective was to verify if a more interactive and engaging type of advertising affects the participants, positively influencing their UPI. In general, by the results obtained through EEG and the EEG engagement index, it was possible to define that a more interactive and engaging type of advertising created a higher UE in the participants. Therefore, it was verified that the advergame increased the level of concentration, surveillance, attention, and engagement of the participants while playing. The graphic correlation between the EEG engagement index variation and the UPI presented in the four-quadrant model showed that most of the time, a more interactive and engaging type of advertising positively influences the UPI. In the case of the advergame, the average UPI was higher than that obtained for the TV commercial (AVG UPI: 4.275 > 3.36), indicating a more substantial influence of the advergame in the post-advertising UPI.

The proposed four-quadrant model is a valid aid for better understanding the relation between the UE and the UPI, allowing to allocate the users in four states: (I) high level of engagement and high UPI, (II) high level of engagement and low UPI, (III) low level of engagement and low UPI, and (IV) low level of engagement and high UPI.

Additionally, the results confirmed some of the advergame advantages identified by Mendiz [4]. Advergames offer high exposure to a brand or product because the participants are in constant contact with them compared to traditional advertising; hence, the participants' levels of engagement are higher when interacting with advergames, maximizing user attention. Similarly, we validated the statement by Goh and Ping [11], proving that users' previous attitudes toward the brand positively influenced the UPI.

We can conclude that it is essential to consider the users' engagement as a parameter to improve advertising campaigns and significantly influence the users, attracting them to a product or brand. Equally important is the right choice of methods (self-report and physiological) implemented to perform this kind of analysis.

Advergames can have both benefits and drawbacks when it comes to user experience and enjoyment. On the positive side, advergames can provide an interactive way for users to engage with a product and enhance their experience. However, the inclusion of ads could also impact the user experience and enjoyment level if the ads are intrusive or irrelevant. For example, if ads are placed in a way that disrupts the gameplay or if they are not pertinent to the player, it can lead to frustration and a negative perception of the overall advergame experience. Thus, the benefits and drawbacks of advergames largely depend on how well the ads are designed and integrated into the game and how they are perceived by the users. In this regard, the analysis of users' perceptions through physiological methods, such as EEG, can offer further considerations to improve the ads' design.

This work presented a use case and validation of EEG-based quantitative indicators in advergames, which can be used alongside self-report methods to minimize biased responses. In this research, the experimental setup included a specific advergame and TV commercial; consequently, the results might differ depending on the advertising to which participants are exposed and the degree of affinity with the product or brand. However, the methodological analysis and the results obtained can be equally considered for similar products; in this sense, there is a research opportunity to explore additional contexts and experimental scenarios to increase the validation value. One potential future research topic could be the EEG analysis of the different design features within the advertising development. Characteristics such as colors, sounds, motion, and forms could be investigated independently to study to which extent these stimuli could influence the users' perceptions. By measuring brain activity, researchers can gain a better understanding of how these factors influence engagement with advertisements. For example, how different colors and musical styles in ads affect attention and memory, or how the use of motion in ads affects the ability to retain information. Likewise, a further research goal is the ecological validity analysis in the real world to understand how well the results of the experiments reflect real-world conditions. In this way, we may be able to develop more effective advertising

strategies that better engage audiences and improve the effectiveness of advertisements. Additionally, this research could also have applications in the fields of psychology and neuroscience, providing valuable insights into the workings of the human brain in relation to advertising.

The methodology presented in this research can be replicated in similar studies interested in deepening the UE analysis. This study is expected to provide practical evidence and theoretical foundations for new approaches to user perception analysis.

**Author Contributions:** Conceptualization, I.A.C.J., F.M. and J.S.G.A.; data curation, I.A.C.J., F.M. and E.C.O.; formal analysis, I.A.C.J and J.S.G.A.; funding acquisition, F.M. and E.V.; investigation, I.A.C.J., S.M., L.U., F.M. and E.C.O.; methodology, I.A.C.J., S.M., L.U. and J.S.G.A.; supervision, F.M. and E.V.; validation, E.C.O., S.M. and L.U.; visualization, S.M., L.U., F.M. and E.V.; writing—original draft, I.A.C.J. and J.S.G.A.; writing—review and editing, I.A.C.J., E.C.O., L.U. and F.M. All authors have read and agreed to the published version of the manuscript.

**Funding:** This research received no external funding.

**Institutional Review Board Statement:** All participants provided written informed consent prior to enrolment in the study, for the correctness, transparency and confidentiality protection, according to art. 13 of the European Regulation n. 679/2016 and Legislative Decree 196/2003, and amended by Legislative Decree 101/2018.

**Informed Consent Statement:** All participants were informed about the experiment and the protection and use of their data. Informed consent was obtained from all subjects involved in the study.

**Data Availability Statement:** Supporting data of this study is not publicly available due to sensitive information that could compromise the privacy policy.

**Conflicts of Interest:** The authors declare no conflict of interest.

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
