# Peer review of "User Engagement Comparison between Advergames and Traditional Advertising Using EEG: Does the User’s Engagement Influence Purchase Intention?"

_electronics, doi:10.3390/electronics12010122_

Round 1
Reviewer 1 Report
The paper is well written, and reading is easy to follow. The correlation between the different fields explored in the paper is well established in the literature presented by the authors. The work is focused in the possibility to enhance the UE throughout a more interactive type of advertising (instead of traditional advertising), and thus influence the user’s purchase intention. To achieve that they compared the UE using EEG in two scenarios. An experiment was created where the users are exposed to two advertisments of the same product while measuring their brain activity with the EEG.
Results show that a more interactive advertisement increased the participants’ EU when compared to a traditional TV commercial. This influenced the user purchase intention.
No major English flaws where detected, just some minor adjustments:
- EEG acronymous is defined in the abstract but not set. This is done only in section 1.
- The same occurs with UE and UPI: multiple redefinitions of the acronymous;
- Please add a reference to sentence in lines 46-48
- Line 142: Is it “on the other hand”? What is being presented is not on the “other hand” of the previous paragraph;
- The hypothesis is only defined in lines 247-249. It should be placed in the introduction.
- Equation 1: can it be written only with Greek letters?
- Figure 4: is it possible to improve quality? When zooming in it looses quality. Possible change of picture format.
The approach presented in the paper is interesting; however, I believe that some more information is required. The stimulus of the brain in the EEG depends, for example, on color, music, motion. To what extent are the advertisements within a similar/different context? Possibly if in the supplementary material the advertisements are presented, this question is answered. Moreover, only with one example can these conclusions be taken? It was just a scenario. Possibly, by adding more scenarios, the validation is of increased value.
Author Response
Following the valuable suggestions made by the reviewer, the revisions to the manuscript and our responses detailed point by point are below. We want to thank the reviewer for the time taken to evaluate our document.
Reviewer
The paper is well written, and reading is easy to follow. The correlation between the different fields explored in the paper is well established in the literature presented by the authors. The work is focused in the possibility to enhance the UE throughout a more interactive type of advertising (instead of traditional advertising), and thus influence the user’s purchase intention. To achieve that they compared the UE using EEG in two scenarios. An experiment was created where the users are exposed to two advertisements of the same product while measuring their brain activity with the EEG.
Results show that a more interactive advertisement increased the participants’ EU when compared to a traditional TV commercial. This influenced the user purchase intention.
No major English flaws where detected, just some minor adjustments:
- EEG acronymous is defined in the abstract but not set. This is done only in section 1.
We have corrected the abstract, adding the complete word and set the EEG acronym from section 1.
- The same occurs with UE and UPI: multiple redefinitions of the acronymous.
Thank you, we have verified and corrected the multiple redefinitions of the acronyms.
- Please add a reference to sentence in lines 46-48
We have added the missing reference.
- Line 142: Is it “on the other hand”? What is being presented is not on the “other hand” of the previous paragraph;
We have corrected the sentence using “likewise” a better word for the reading context.
- The hypothesis is only defined in lines 247-249. It should be placed in the introduction.
We have included the hypothesis in the introduction section.
- Equation 1: can it be written only with Greek letters?
Yes, we have written the equation using only Greek letters
- Figure 4: is it possible to improve quality? When zooming in it looses quality. Possible change of picture format.
Yes, we have included a better-quality image
- The approach presented in the paper is interesting; however, I believe that some more information is required. The stimulus of the brain in the EEG depends, for example, on color, music, motion. To what extent are the advertisements within a similar/different context? Possibly if in the supplementary material the advertisements are presented, this question is answered. Moreover, only with one example can these conclusions be taken? It was just a scenario. Possibly, by adding more scenarios, the validation is of increased value.
Thank you for your valuable observations. We have addressed your observations in the conclusion sections as potential future research topic, as follows:
“In this research, the experimental setup included a specific Advergame and TV commercial; consequently, the results might differ depending on the advertising to which participants are exposed and the degree of affinity with the product or brand. Yet, the methodological analysis and the results obtained can be equally considered for similar products, in this sense, there is an open research opportunity to explore additional contexts and experimental scenarios.
One potential future research topic could be the EEG analysis of the different design features within the advertising development; characteristics such as, colors, sounds, motion, and forms, could be investigated independently to study to which extent these stimuli could influence the users’ perceptions. By measuring brain activity, researchers can gain a better understanding of how these factors influence engagement with advertisements. For example, how different colors and musical styles in ads affect attention and memory, or how the use of motion in ads affects the ability to retain information. Likewise, a further research goal is the ecological validity analysis in real-world, to understand how well the results of the experiments reflect real-world conditions. In this way, we may be able to develop more effective advertising strategies that better engage audiences and improve the effectiveness of advertisements. Additionally, this research could also have applications in the fields of psychology and neuroscience, providing valuable insights into the workings of the human brain in relation to advertising.”
Regarding the description of the advertisement, in section 3.2, we present a description of the advergame and tv commercial, however due to copyright reasons we are not allowed to publish any image of the advertising campaign. We apologize for the inconvenience.

Reviewer 2 Report
General comments
----------------
Hypothesis: "A more interactive type of advertising impacts the user perception and engagement, positively influencing the purchase intention."
Contribution: Probably first in using EEG for measuring the UE influence on UPI, which is valuable in itself. The work has a value also on the theoretical side by introducing a model that connects UE with UPI (the quadrant model).
The manuscript is very well structured and very well written.
There is one major flaw in the experiment design though: the order in which participants were exposed to the two conditions were not randomized. The ~20 participants were exposed to the advergame first, _then_ the TV commercial, in all instances. There is a huge probability that fatigue influenced the results and the authors do not seem to be aware of this: All participants could potentially be a bit less engaged in the TV commercial condition simply because of this. If the authors would have a compelling argument for why this risk is not there, the study would be much more trustworthy but as it stands now, the result is questionable.
A minor flaw is the lack of reflection on ecological validity in the real world. Can we really be sure that people would be more likely to go off bying Coke after playing advergames if compared to watching TV commercials?
The term "questioning" is used in the manuscript in different parts (e.g. in table 1) without precise explanation to whether this refers to written (e.g. through online or paper questionnaires), oral (e.g. through interviews or focus group sessions), or other forms of question-based data elicitaton. More precision is relevant because the type of "questioning" method comes with different strengths and drawbacks.
I miss a discussion on the potental User Experience (enjoyment) level decrease due to introduction of advertisements in the games. Maybe games without ads will be chosen more frequently than advergames because of this, decreasing the target group?
Detailed comments
-----------------
p5 r206: I have the feeling that "product price" is missing in this list of UPI factors?
p6 Fig 1: Hard to interpret.
p7 formula: why large space between "Thet" and "(theta sign)" and why the prime ("'")?
p8 r297 "if it increases, a higher UE is expected" is probably better formulated as "if it increases, this is interpreted as a higher level of UE"
p8 r313 A photo or a more detailed description of the experiment environment would be valuable as things like the distance to the computer screen and the size of it might impact the level of the UE, or not?
p9 r320 What was the "product selected for the study"?
p9 r326 A screen shot of the Advergame would be nice.
p9 r326 Why not randomized order between the Advergame and the TV commercial to counteract potential fatigue that could affect UE?
p11 Fig 7: It would be more logical to display the three UPI conditions in the order they were measured so original UPI to the left instead of to the right.
p11 The claim that "The outcomes show that a more engaging type of advertising significantly 384 influences the UE" needs to come _after_ the ANOVA details a few rows later.
p13 Wouldn't it be relevant to discuss the difference in absolute number of points in the rightmost halves of figure 9 and 10? The fact that Advergame has a positive impact on UPI for 13 participants while TV commercial has this only for 7?
p14 There something wrong in this sentence: "users that have a positive attitude toward the brand or product advertised in games have a more significant influence on purchase intention"
Author Response
Following the valuable suggestions made by the reviewer, the revisions to the manuscript and our responses detailed point by point are below. We want to thank the reviewer for the time taken to evaluate our document.
Reviewer
General comments
Hypothesis: "A more interactive type of advertising impacts the user perception and engagement, positively influencing the purchase intention."
Contribution: Probably first in using EEG for measuring the UE influence on UPI, which is valuable in itself. The work has a value also on the theoretical side by introducing a model that connects UE with UPI (the quadrant model).
The manuscript is very well structured and very well written.
- There is one major flaw in the experiment design though: the order in which participants were exposed to the two conditions were not randomized. The ~20 participants were exposed to the advergame first, then the TV commercial, in all instances. There is a huge probability that fatigue influenced the results and the authors do not seem to be aware of this: All participants could potentially be a bit less engaged in the TV commercial condition simply because of this. If the authors would have a compelling argument for why this risk is not there, the study would be much more trustworthy but as it stands now, the result is questionable.
Thank you for your valuable observation; we want to clarify that the execution of the experiment was randomized for the participants, half of them were first assigned to play the advergame and then watch the tv commercial, while the other half first watched the tv commercial and then proceeded with the advergame. In Figure 4, we depicted at the same “execution level” both the stimuli (advergame and tv commercial) as parallel activities to highlight the fact that these activities were carried out concurrently (one participant watching the tv commercial and another playing the advergame), rather than one after the other. We realized, however, that the experimental setup (figure 4) and the description below was ambiguous, and it could lead to misinterpretations.
We apologize for the misunderstanding. We have updated the description and clarified the experiment in a better way.
- A minor flaw is the lack of reflection on ecological validity in the real world. Can we really be sure that people would be more likely to go off bying Coke after playing advergames if compared to watching TV commercials?
The ecological validation in a real-world setting is beyond the scope of this study since, in this research, we are mainly interested in exploring EEG as a means to analyze User Engagement between advergames and traditional advertising. However, your observation is pertinent, and we have included it in the conclusions section as part of future potential studies.
- The term "questioning" is used in the manuscript in different parts (e.g. in table 1) without precise explanation to whether this refers to written (e.g. through online or paper questionnaires), oral (e.g. through interviews or focus group sessions), or other forms of question-based data elicitation. More precision is relevant because the type of "questioning" method comes with different strengths and drawbacks.
Thanks for your observation. We have changed the word “questioning” to a more appropriate term “self-report”, and added a paragraph in section 2.1 to provide a more detailed explanation on this regard. Also, we have specified in table 1 the type of self-report method used in each study.
“Other studies on UE analysis in video games are listed in Table 1. Most of these studies, however, relied only on self-report methods, such as questionnaires, inter-views, surveys, and scales, in which participants are asked to answer questions or pro-vide responses to prompts that are designed to collect data on a particular topic or construct. The data collected through self-reported methods can be subjective and may be influenced by various factors, such as the participants' memory, motivation, and willingness to reveal personal information. Despite these limitations, self-reports have been widely used as a method for UE analysis, and as a consequence, this could lead to biased outcomes due to the nature of the responses. In this sense, self-reports could be compared and enhanced with other experimental approaches that do not depend entirely on participants' responses, such as the case of EEG.”
- I miss a discussion on the potential User Experience (enjoyment) level decrease due to introduction of advertisements in the games. Maybe games without ads will be chosen more frequently than advergames because of this, decreasing the target group?
We have addressed your suggestion in the discussion session as follows:
“Advergames can have both benefits and drawbacks when it comes to user experience and enjoyment. On the positive side, advergames can provide an interactive way for users to engage with a product and enhance their experience. However, the inclusion of ads could also impact the user experience and enjoyment level if the ads are intrusive or irrelevant. For example, if ads are placed in a way that disrupts the gameplay or if they are not pertinent to the player, it can lead to frustration and a negative perception of the overall advergame experience. Thus, the benefits and draw-backs of advergames largely depend on how well the ads are designed and integrated into the game and how they are perceived by the users. On this regard, the analysis of users’ perceptions through physiological methods, such as EEG, can offer further considerations to improve the ads’ design.”
Detailed comments
- p5 r206: I have the feeling that "product price" is missing in this list of UPI factors?
Yes, you are right, we have included it in the description.
- p6 Fig 1: Hard to interpret.
We have included an introduction to the figure to complement and ease its interpretation, as follows:
“The relationship between videogames and advertising is complex and multifaceted, with each industry influencing and benefiting from the other in various ways; in the intersection between there are the advergames (Figure 1), which provide a unique and highly engaging platform for advertisers to reach consumers, with in-game ads and sponsored content offering new and innovative ways for brands to connect with their audience and promote a product or service through the use of interactive and engaging gameplay. This strategy relies on user engagement to be successful, with the level of engagement (UE) influencing the effectiveness of the advertising and the likelihood of a user making a purchase (UPI).
The interaction between these three contexts (videogames, advertising, and advergames) and the UE influence on the UPI frames this research.”
- p7 formula: why large space between "Thet" and "(theta sign)" and why the prime ("'")?
We think it was a visualization mistake, we have checked and verify that the formula is correctly written.
- p8 r297 "if it increases, a higher UE is expected" is probably better formulated as "if it increases, this is interpreted as a higher level of UE"
Thank you for your observation, we have updated the manuscript according to your suggestion.
- p8 r313 A photo or a more detailed description of the experiment environment would be valuable as things like the distance to the computer screen and the size of it might impact the level of the UE, or not?
According to your observation, we have included a more detailed description of the experiment environment, as follows:
“The experiment was displayed on a 40-inch curved display with an effective viewing area of 884.74mm (horizontal) and 497.66mm (vertical) at a 3000mm radius curvature with a pixel density of 110 per inch. The participants sat in front of the display, ap-proximately 800mm (± 50mm). All the participants (one at a time) executed the test in the same room. This monitor was selected as studies [52,53] showed that consider a large field of view of a participant can greatly enhance their experience, and with the help of its curvature and high pixel density it can increase immersion.”
In the file attached, for your information, we have included a couple of photos of the experiment during resting condition, that we use only for internal communication purposes, therefore there are not included in the paper.
- p9 r320 What was the "product selected for the study"?
The selected product belongs to a well-known brand of carbonated soft drinks, specifically Pepsi.
- p9 r326 A screen shot of the Advergame would be nice.
Unfortunately, due to copyright policies we are not able to publish any image of the advergame or tv commercial, we apologize for the inconvenience.
- p9 r326 Why not randomized order between the Advergame and the TV commercial to counteract potential fatigue that could affect UE?
As previously explained the order was actually randomized. We have described better the experimental setup (section 3.2) as follows:
“Half of the participants were randomly assigned first to play an Advergame and then proceed with a TV commercial, while the other half initially watched the TV commercial and then continued with the Advergame activity. In both cases, the participants were asked to play the Advergame for 4 minutes and exposed to a TV commercial for 3 minutes while measuring their brain activity. Between one activity and the other, the participants were invited to relax for 2 minutes.”
- p11 Fig 7: It would be more logical to display the three UPI conditions in the order they were measured so original UPI to the left instead of to the right.
We have updated the figure to display the three UPI conditions in the order they were measured.
- p11 The claim that "The outcomes show that a more engaging type of advertising significantly 384 influences the UE" needs to come _after_ the ANOVA details a few rows later.
We have updated the paragraph according to your suggestion.
- p13 Wouldn't it be relevant to discuss the difference in absolute number of points in the rightmost halves of figure 9 and 10? The fact that Advergame has a positive impact on UPI for 13 participants while TV commercial has this only for 7?
Thanks for your suggestion, we have included your observation at the end of section 5, as follows:
“Finally, the contrast in the total number of points in the rightmost halves of both Figures 9 and 10 shows that the advergame had a positive impact on UPI for thirteen participants (65%) while the TV commercial only for seven (35%); this implies that apart from the UE elicitation, the advergame was most effective at increasing participants’ purchasing interest. By playing the advergame, users were exposed to the product in a pleasant and enjoyable way, enhancing their interest in it and increasing their likelihood of considering buying it. Overall, advergames can be an effective marketing strategy to inform customers about the product and its characteristics, improve users’ understanding of the product and boost their desire to purchase it.”
- p14 There something wrong in this sentence: "users that have a positive attitude toward the brand or product advertised in games have a more significant influence on purchase intention"
We have corrected the sentence.
